# A Framework to Build a Big Data Ecosystem Oriented to the Collaborative Networked Organization

Jorge-Arturo Hernandez-Almazan [1], Ricardo Chalmeta [2,*], Ramón Ventura Roque-Hernández [3] and Rubén Machucho-Cadena [1]

1 Department of Knowledge-Driven Systems, Universidad Politécnica de Victoria, Ciudad Victoria 87138, Mexico
2 Systems Integration and Re-Engineering Group (IRIS), Lenguajes y Sistemas Informáticos Departament, Universitat Jaume I, 12006 Castelló de la Plana, Spain
3 Facultad de Comercio, Administración y Ciencias Sociales, Universidad Autónoma de Tamaulipas, Nuevo Laredo 88000, Mexico
* Correspondence: rchalmet@uji.es

**Abstract:** A Collaborative Networked Organization (CNO) is a set of entities that operate in heterogeneous contexts and aim to collaborate to take advantage of a business opportunity or solve a problem. Big data allows CNOs to be more competitive by improving their strategy, management and business processes. To support the development of big data ecosystems in CNOs, several frameworks have been reported in the literature. However, these frameworks limit their application to a specific CNO manifestation and cannot conduct intelligent processing of big data to support decision making at the CNO. This paper makes two main contributions: (1) the proposal of a metaframework to analyze existing and future frameworks for the development of big data ecosystems in CNOs and (2) to show the Collaborative Networked Organizations–big data (CNO-BD) framework, which includes guidelines, tools, techniques, conceptual solutions and good practices for the building of a big data ecosystem in different kinds of Collaborative Networked Organizations, overcoming the weaknesses of previous issues. The CNO-BD framework consists of seven dimensions: levels, approaches, data fusion, interoperability, data sources, big data assurance and programmable modules. The framework was validated through expert assessment and a case study.

**Keywords:** data analytics; interoperability; knowledge management; knowledge discovery; information systems; enterprise networks; virtual enterprises

## 1. Introduction

A Collaborative Networked Organization (CNO) is composed of companies and/or people in nonhomogeneous environments in terms of activities, culture, social capital and goals, independently governed and geographically distributed, where the participants share their competencies to collaboratively address a temporary or long-term common or compatible objective [1]. The goals of CNOs are: (1) to maximize the overall network profit; (2) to achieve the local objectives and develop the strategies of each member of the CNO; and (3) to withstand market fluctuations while minimizing the costs of the total network and each member of the CNO [2].

CNOs show high potential not only in terms of survival capability, but also value creation due to their ability to cope with innovation needs, uncertainty, mass customization and fierce competition [3]. It is expected that, by working together, companies can receive the following benefits: reaching (apparent) larger dimensions, through their access to new/wider markets and new knowledge; information sharing of risks and resources; creation of a collaborative environment for innovation, through the combination of synergy, competencies, culture and experiences; the smooth integration of customers in the product development and innovation process; and linking of complementary skills and capacities,

which allows each entity to focus on its core competencies while retaining a high level of agility [1].

During their activities, CNOs generate and store large heterogeneous datasets [4], cataloged as big data [5]. Therefore, CNOs can increase their productivity and efficiency through a big data ecosystem, which allows them to collect, store and process data from different sources and in different formats, as well as visualize and disseminate information for decision making [6].

Building a big data ecosystem in a CNO is a technically challenging task [7] because it should offer the following capabilities: big data storage, big data processing, big data orchestration, big data assistance, big data interfacing and big data deployment [8]. In addition to this, a big data ecosystem can significantly alter the business processes of the CNO because it can lead to changes in the decision making of top management based on the insights obtained [9]. Consequently, the development and implementation of a big data ecosystem in a CNO is a very complex task that includes not only technological aspects, but also strategy, management, business process and human resources issues.

To support practitioners, several frameworks have been proposed in the literature for big data ecosystems for the development and implementation of CNOs. These frameworks are mainly focused on addressing the terminology, characteristics and elements of big data ecosystems. Although, these frameworks are very interesting contributions, in their present state there are some issues that limit their effectiveness: (1) none of the existing frameworks consider all the necessary dimensions for the development of a big data ecosystem, and therefore none can provide holistic support; (2) their application is limited to a specific CNO manifestation, and therefore their scope is very restricted; (3) none consider certain important technical issues such as guides for data fusion, the combination of heterogeneous data, interoperability improvement or the selection of the appropriated big data analysis techniques; and (4) they are focused only on the technical issues of big data ecosystem development, without considering the impact on the CNO configuration, human resources and business processes.

To overcome these issues, this paper proposes a new framework called the CNO-BD framework. This supports the development and implementation of a big data ecosystem in CNOs. Moreover, it takes into account the main characteristics of a big data ecosystem and avoids all the analyzed frameworks' weaknesses described above. Therefore, unlike the existing frameworks, the CNO-BD framework provides holistic support, can be used across different CNO manifestations, takes into account technical issues not considered in other frameworks and provides support to the CNO and its management.

The remainder of this paper is organized as follows: A literature review is provided in Section 2. Section 3 proposes the CNO-BD framework. Section 4 outlines the two methods used to improve and validate the CNO-BD framework. Section 5 summarizes this study's findings and limitations and gives suggestions for future work. Finally, Section 6 presents the conclusions.

## 2. Literature Review

### 2.1. Collaborative Networked Organization (CNO)

A CNO is a set of entities that operate in heterogeneous contexts and aim to collaborate to take advantage of an opportunity or solve a problem. There are many forms of collaboration. According to the taxonomy developed by [10], the different forms of CNO can be organized into:

(a) Goal-oriented continuous production-driven networks: These are CNOs that remain stable for a long period of time with well-defined roles. Examples of this kind of CNO are the supply chains (SC) and collaborative smart grids of all business sectors and Virtual Governments (VG), i.e., an alliance of governmental organizations (e.g., city hall, tax office, cadastre office and civil infrastructure office) that combine their services through the use of computer networks to provide integrated services to citizens through a common front-end.

(b) Goal-oriented grasping opportunity-driven networks: These are CNOs composed of groups of independent organizations or individuals sharing skills and resources, dynamically created in response to some business opportunity within a limited time window. This kind of CNO can be classified as a Virtual Organization (VO), Dynamic Virtual Organization (DVO)/virtual enterprises (VE), Extended Enterprise (EEs) and Virtual Teams (VT). Examples of each one of these CNOs can be found in [11].

(c) Long-term breeding environment networks (VBE), which are a group of organizations and supporting institutions committed to a long-term cooperation agreement, complying with common operation principles and infrastructures, with the main goal of increasing their preparedness for the rapid configuration of temporary alliances for collaboration in potential VOs. In the literature, four types of VBEs have been identified: industry clusters, composed of groups of inter-related industries geographically concentrated and inter-connected by the flow of goods and services, which drive wealth creation in a region, primarily through the export of goods and services; business ecosystems, a set of organizations that have resources and act together cooperatively or competitively to form a unique independent system (examples of these CNOs are platform-based ecosystems (Apple or Google), start-up ecosystems (Canvas) or mobility ecosystems (Uber)); disaster rescue networks, a strategic alliance of governmental/nongovernmental organizations specialized in rescue operations in case of disasters; virtual laboratories (VL), such as the project carried out by the Ministry of Education of India to provide remote access for students at all levels to laboratories in various disciplines of science and engineering, to perform the experiments remotely.

(d) Long-term professional virtual communities (PVC): These are similar to VO breeding environment networks but composed of human professionals (self-employees, freelancers, etc.) that form a long-term strategic alliance to ensure they are prepared to react quickly to business opportunities through the dynamic creation of temporary VTs. An example of a PVC is the European Society of Concurrent Enterprising Network, a nonprofit association bringing together academics, researchers and industry to exchange ideas, views, practices and the latest research and developments in the field of concurrent enterprising.

In order to successfully operate, members of the CNO need to easily interchange data and information. This is known as interoperability [12], which is important not only for the individual enterprise, but also for the new business structures that are emerging as CNOs [13].

*2.2. Big Data in the CNO*

Big data is inevitable in the current economic, scientific and industrial domains [14,15], being an effective tool for achieving the Sustainable Development Goals (SDGs) [16]. Big data is changing infrastructures and implementation strategies of Information and Communication Technologies (ICT) in the CNO [17]. Although big data has multiple definitions in the literature, it is characterized by six Vs: volume, velocity, variety, value, veracity and variability [18]. Traditional technologies and platforms are unable to meet the new requirements [19], and therefore new technologies are required to access, collect and process big data [20], such as machine learning [21].

In CNO, soft and hard data are mass generated. Hard data are generated by nonhuman entities, whereas soft data are generated by a human in natural language [22]. Additionally, the data can be classified according to their frequency of use as cold data, which are accessed sporadically, and hot data, which are accessed frequently.

Each participant of the CNO has different data sources, categorized, in accordance with their nature, as structured, unstructured [23] or semi-structured [24]. It is necessary to transform these heterogeneous data into useful knowledge and share it among the CNO's participants to optimize decision making [25]. In addition, another problem is that multiple data sources can cause data redundancy. This hinders the obtention of valuable information from the CNO. Big data analytics can reduce both uncertainties in CNO decisions and deal

with heterogeneous CNO data. Therefore, using of the big data generated in the processes and interactions of the CNO is challenging.

### 2.3. Big Data Frameworks Oriented to a CNO

A framework organizes scattered theoretical ideas and establishes good practices according to the dimensions considered [26]. The framework integrates evaluation strategies [27], practices, tools and actors based on the literature [28].

Ten frameworks have been proposed to guide during the development of a big data ecosystem in CNOs. Based on a review of these frameworks, a metaframework was developed in this study. This metaframework identifies six dimensions and several sub-dimensions that a proper big data framework for CNOs should fulfill. The six dimensions and their sub-dimensions (showed in brackets) are: levels (based on architecture), approaches (data alignment, compression, centralized and decentralized), data fusion (alternative technique based on context), interoperability (mechanism to remove incompatibilities), data sources (structured, semi-structured and unstructured) and big data assurance (six Vs). By mapping existing frameworks with the metaframework, it is possible to easily identify the main features of every framework. Table 1 shows the dimensions and sub-dimensions considered in the ten existing frameworks that manage big data ecosystems oriented to a CNO.

**Table 1.** Dimensions and sub-dimensions considered in the revised frameworks.

| Framework | Levels: Based on Architecture | Approaches: Data Alignment | Approaches: Compression | Approaches: Centralized | Approaches: Decentralized | Data Fusion: Alternative Technique Based on Context | Inter Operability: Mechanism to Remove Incompatibilities | Data Sources: Structured | Data Sources: Semi-Structured | Data Sources: Unstructured | Big Data Assurance: 6 Vs |
|---|---|---|---|---|---|---|---|---|---|---|---|
| Li et al., 2015 [29] | • | • |  | • | • | • | • | • | • |  |  |
| Ilie-Zudor et al., 2015 [30] | • |  |  | • | • |  | • | • | • |  |  |
| Wang et al., 2016 [31] | • |  |  |  |  |  | • | • |  |  |  |
| Klievink et al., 2016 [32] |  | • |  |  | • |  | • | • | • | • |  |
| Chen et al., 2017 [33] | • |  |  | • |  |  |  | • | • | • | • |
| Intezari et al., 2017 [34] |  |  |  | • |  |  |  | • |  | • | • |
| Sadic et al., 2018 [35] | • |  |  | • |  | • |  | • |  |  |  |
| Brisimi et al., 2018 [36] |  | • |  | • | • |  | • | • | • |  | • |
| Yi et al., 2018 [37] |  |  | • | • | • |  | • | • |  |  |  |
| Orenga-Roglá et al., 2018 [38] | • | • |  | • |  |  | • | • | • | • | • |

Bullet points shows the dimensions and subdimensions cover by each framework.

The previously mentioned frameworks allow the CNO to take advantage of some benefits of big data and related technologies. All of them have solid bases. However, they also have several weaknesses that need to be addressed:

- None of the existing frameworks include all dimensions. Therefore, they lack the fundamental aspects for a big data ecosystem oriented to a CNO.
- Frameworks must complement the levels dimension, utilizing the sub-dimension explicit in the design of the CNO. Thus, they lack a graphic model that shows levels in any manifestation of CNO.
- Frameworks must extend the approaches dimension through the Inclusive sub-dimension, which can improve the processing of large datasets in the CNO.
- Frameworks must extend the data fusion dimension through the sub-dimension techniques guide to guide the combination of heterogeneous data [39–47].

- Frameworks do not indicate the degree of interoperability among the participants. Therefore, frameworks must complement the interoperability dimension through the sub-dimension maturity model.
- Frameworks do not include alternatives for big data analysis techniques. Therefore, frameworks must extend the big data assurance dimension using the sub-dimension techniques guide.
- Frameworks should add the programmable modules dimension, which allows the intelligent processing of big data in the CNO.
- Each framework limits its orientation to a specific CNO's manifestation. Those of Li et al. [29], Ilie-Zudor et al. [30] and Wang et al. [31] are oriented to SC; Klievink et al.'s [32] is oriented to VG; Chen et al.'s [33] is oriented to PVC; Intezari and Gressel's [34] is oriented to VBE; Sadic et al.'s [35] is oriented to VE; Brisimi et al.'s [36] is oriented to VO; Yi et al.'s [37] is oriented to VL; and Orenga-Roglá and Chalmeta's [38] is oriented to EE.

Considering the weaknesses mentioned before, there is a need for a new framework that (1) manages big data ecosystems oriented to a CNO; (2) incorporates and effectively uses big data generated in the CNO to improve its performance; and (3) avoids all identified weaknesses.

## 3. Proposed Framework

In this context, the CNO-BD framework that supports the building of a big data ecosystem oriented to the CNO is proposed. It focuses on (1) the acquisition and intelligent processing of big data and (2) the establishment of good practices, techniques, tools and guidelines that support the CNO big data ecosystem.

The first version of the CNO-BD framework was developed considering the scientific and academic advances in this line of research, and the experience acquired by the authors in the development and implementation of big data ecosystems. We used research from domains of collaborative organizations, interoperability, information systems and data fusion to design the CNO-BD framework, particularly that on big data. Thus, we conceptualized the use of big data within the CNO. The next version of this framework was evaluated by five experts in big data from different companies/institutions. They provided observations and suggestions that allowed us to improve it. The penultimate version of the framework was validated through a case study in a CNO in the education sector in Mexico. The personnel involved in the project provided feedback to improve it. Thus, the final version of the CNO-BD framework was obtained.

In the CNO-BD framework, it was noticed that a new dimension that was not being previously considered by the other frameworks was needed. This dimension is programmable modules, which allows big data to be intelligently processed in the CNO. Therefore, the CNO-BD framework is composed of seven dimensions with their respective sub-dimensions (Figure 1): levels dimension, data sources dimension, approaches dimension, data fusion dimension, big data assurance dimension, programmable modules dimension and interoperability dimension. Highlighted in bold are (1) the sub-dimensions that are proposed to be added to each dimension, which extends the necessary big data aspects in the CNO-BD framework and (2) the seventh dimension, called programmable modules, which process big data intelligently in the CNO.

The dimensions are interrelated. The levels dimension considers the different levels of the CNO. These provide data that are from different data sources. These data are collected by the approaches tools and methods, and their heterogeneity is solved by the data fusion techniques. Then, big data assurance provides different methods than can be used by the programmable modules to generate the proper information and knowledge for decision making at the different levels of the CNO. Finally, interoperability provides support in data extraction from the different levels and their return in the form of information and knowledge.

All dimensions and sub-dimensions, except those indicated in bold, are considered in at least one of the ten frameworks reviewed. However, none of the existing frameworks include all of them.

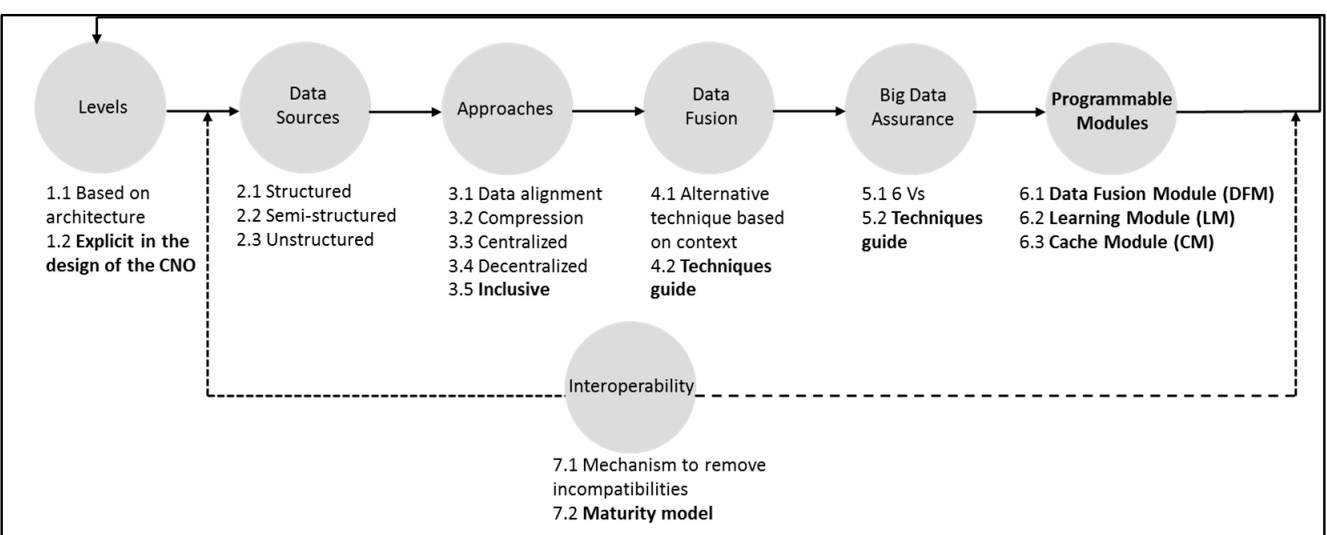

**Figure 1.** The CNO-BD framework's dimensions and sub-dimensions.

### 3.1. Levels Dimension

The CNO-BD framework extends this dimension through the sub-dimension explicit in the design of the CNO, which shows three explicit levels in the design of a generic architecture oriented to the CNO. Each CNO's manifestation can use the proposed design as a starting point and adapt it to its needs. Additionally, the levels act as a guide to use components from other dimensions. Figure 2 shows the levels that compose the CNO's architecture, which are:

- The top level: A CNO's participant is located here. It has the greatest processing resources and accessibility to the Internet. Participants' requests are attended by a Data Processing Center (DPC). The DPC processes structured, semi-structured and/or unstructured datasets using two options: (1) the local environment: Hadoop, relational database management system, etc.; and (2) services in the cloud: Amazon Web Services (AWS), Microsoft Azure, etc. The DPC can access content in social networks, the information in the cloud, and so on.

- The middle level: From 1 to n participants are located here. They have mid-range infrastructures and are close to the DPC and function as intermediate nodes between the extreme levels. Additionally, they can process and respond to some bottom-level requests without consulting the DPC; see the approaches dimension. This level is linked at high speed with the DPC to allow the interoperability and fusion of global data in the shortest possible time. The DFM, LM and CM components are explained in the programmable modules dimension.

- The bottom level: From 0 to n participants are located here. They have low-end infrastructures or are further from the DPC compared to the middle level. They are served by multiple middle-level participants. The bottom level provides data to enrich the analysis carried out in the higher levels. However, it can store information processed and replicated by the middle level, such as the most popular information in the CNO. Just like the bottom level, the middle level can also store the most popular information.

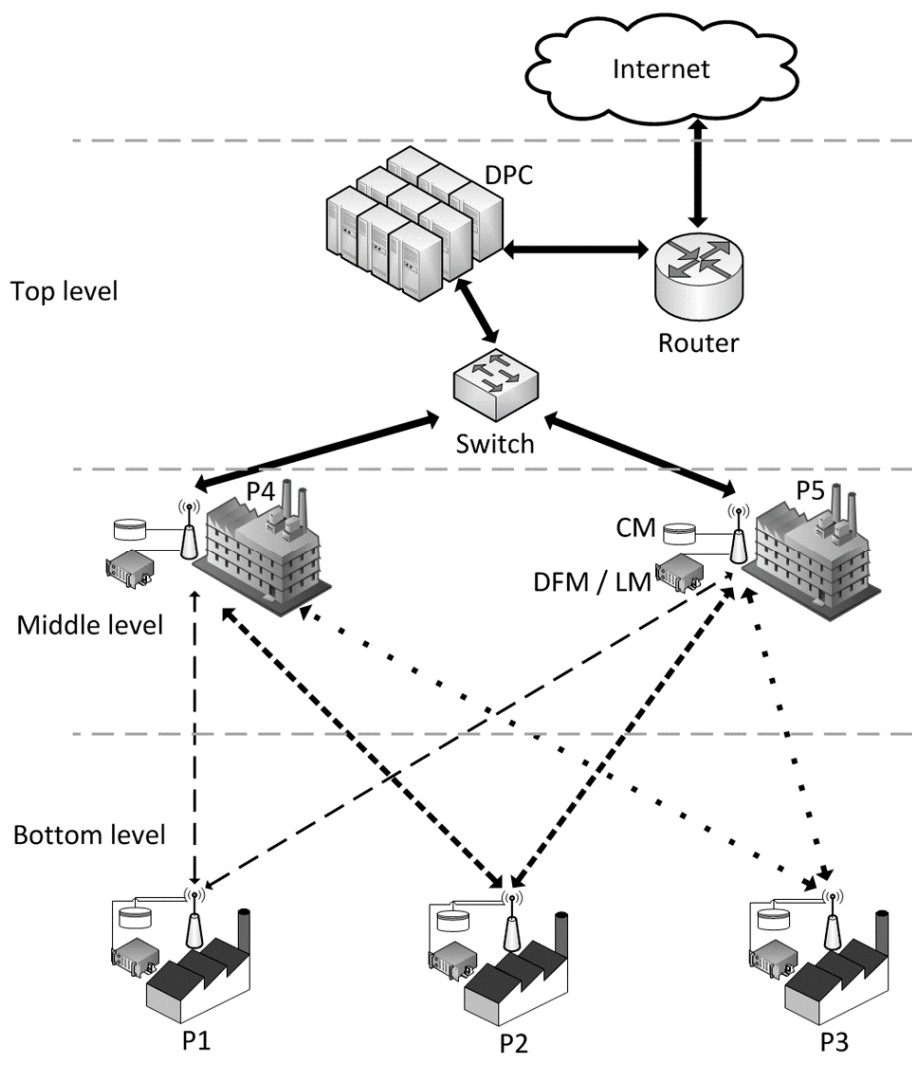

**Figure 2.** CNO's architecture levels.

### 3.2. Data Sources Dimension

This dimension aims to emphasize the inclusion of the types of data sources in a big data ecosystem, which are: (1) structured data: based on a logical model to represent the data; (2) semi-structured data: may have a specific internal structure, but not compatible with a relational database; and (3) unstructured data: no specific pattern. Table 1 shows that structured data sources are considered in all the frameworks studied. In contrast, semi-structured data are less popular, and unstructured data are minimal. Therefore, the structured data source is the most frequently used in big data ecosystems. Nevertheless, the worldwide trend is an exponential growth in the amount of unstructured data, which will make them the most popular. Additionally, it can be combined with the other two types of data sources. Thus, the unstructured data category represents a big opportunity to extract valuable information.

This dimension proposes that a big data ecosystem includes at least two different types of data sources that maximize the value extracted from big data and complies with the characteristics of variety and variability. These two Vs are less frequently considered in current big data ecosystems.

### 3.3. Approaches Dimension

The approaches dimension has two objectives: (1) to preprocess local data of each CNO participant before being transferred to the higher level and (2) support the intelligent processing of big data. Thus, it reduces the use of resources. This dimension considers five approaches, which are: data alignment, compression, centralized, decentralized and inclusive. The approaches are used in the components of the programmable modules dimension. Each approach uses different techniques according to CNO's architecture levels; see the data fusion dimension. Table 2 shows the relationship between the approaches with the levels mentioned before.

- Data alignment: This reduces errors in the data to be processed in the upper level. Each CNO's participant must transform their data from its specific orientation to collaborative orientation before data fusion. This challenge can be overcome by using methods for data alignments, such as the primary key, indexes, heuristic matching, Monte Carlo simulations or cross-correlation analysis methods.
- Compression: This reduces the use of resources in the CNO, such as data transfer in the network, disk space, DPC processing and power consumption. The CNO may use lossy compression and lossless compression methods.
- Centralized: This sub-dimension combines a CNO's global data sources in the DPC. Thus, a participant may request information from the DPC when: (1) local information is not enough or (2) the middle level does not satisfy the request. This approach processes data in two ways: streaming processing to obtain results in real time and batch processing for extended processing.
- Decentralized: This allows the participant to take advantage of the local and/or the most popular information of the CNO; the latter is replicated from the level immediately superior to the participant. Thus, the participant can make decisions without consulting their superior level. Therefore, when all participants need to send requests to the DPC (centralized), the decentralized approach can consume fewer resources, including during networking and processing.
- Inclusive: The centralized approach avoids robust local infrastructure for each participant because it moves computation to the top level. However, big data generated in the CNO can overwhelm this fully centralized approach. The CNO-BD framework extends this dimension through the inclusive sub-dimension to adaptively choose the processing, as follows: (1) local in each participant (decentralized), (2) distributed between local and other participants, (3) global in the DPC (centralized) and/or (4) simultaneous processing through the three previous types. This approach considers the content of the data and their relationships. Thus, the data processing is distributed among the participants which may balance the workload with many requests for large datasets. The distribution of data processing has two characteristics: (1) the data can be transferred to a participant from both DPC or requesting node and (2) data transfer can be avoided by using the most popular CNO information. As a result, the DPC only processes information that is not accessible at the two lower levels. The inclusive approach uses techniques and components of the data fusion dimension, the big data assurance dimension and the programmable modules dimension. This synergy allows the intelligent processing of big data.

### 3.4. Data Fusion Dimension

The CNO-BD framework extends this dimension through the sub-dimension techniques guide, which allows the big data ecosystem to be extended. This sub-dimension provides techniques for the fusion of heterogeneous data and examples of their use according to the levels; see Table 3. The techniques can be adapted to the context and needs of each CNO's manifestation and are used in the approaches dimension and the programmable modules dimension. Although these techniques are implemented mainly in the top level, the two other levels of the CNO also apply them. Data fusion can contribute to the CNO by (1) reducing the number of messages between participants—and,

therefore, local data can be combined with the routing time to transfer only the resulting combination—and (2) supporting new workflows or new software interfaces of recently incorporated participants.

**Table 2.** Approaches mapped in CNO's architecture levels.

| Approaches | CNO's Architecture Levels | | |
|:---:|:---:|:---:|:---:|
| | **Bottom** | **Middle** | **Top** |
| Data alignment | ● | ● | |
| Compression | ● | ● | |
| Centralized | | | ● |
| Decentralized | ● | ● | |
| Inclusive | ● | ● | ● |

Bullet points shows the levels cover by each approach.

**Table 3.** Use of data fusion techniques oriented to the CNO.

| Lit. | Technique/Level | Description |
|:---:|:---:|:---|
| Atrsaei et al., 2016 [39] | Kalman filter/ B, M and T | This technique estimates the current state of a system. It includes the fusion of current and past data collected from the system itself and can be used in streaming processing and/or batch processing applications. Therefore, it can be used in the centralized and decentralized approaches of the CNO-BD framework. Additionally, it allows the participants to recalculate corrupt or incomplete results. |
| Alvarez et al., 2018 [40] | Dempster–Shafer theory/B, M and T | This technique determines epistemic uncertainty and stochastic uncertainty. These are represented in parallel or individually by a probability box. Each participant that implements it can provide information to build a belief network. For example, some participants provide information to identify unitary elements, while other participants provide information to identify classes of elements. |
| Guo et al., 2017 [41] | Fuzzy set theory/ B, M and T | This technique manages the lack of clarity in the data. It works with partial or vague data sources that are fuzzified with a membership function. This can convert the values of a CNO into qualitative attributes that can be evaluated with heuristic rules. This technique can be used in the transportation of products of an SC since decisions are made based on uncertainty and inaccurate data. |
| Dell'Orco et al., 2017 [42] | Possibility theory/B and M | Just like fuzzy set theory, possibility theory can use multisource data uncertainty; uncertainty is present in imprecise or vague knowledge. This technique considers human perception and behavior, and the distribution of information. For example, a CNO of the agrifood sector makes decisions based on humans' perceptions, such as the quality control of the product. |
| Chen et al., 2018 [43] | Rough set theory/ B, M and T | This technique can fuse inaccurate data by considering its granularity (internal structure), and does not require extra information. It avoids dimensional drawbacks, reduces the time factor, and can be implemented in the CNO to (1) preprocess and mine incoming data from multiple sources; (2) identify the data characteristics; (3) remove redundant information; and (4) discover patterns in the datasets. |
| Zampieri et al., 2018 [44] | Kernels/T | Kernels are similar measurements between data pairs. They can represent nonhomogeneous datasets as kernel arrays. Thus, different types of CNO data and/or formats can be transformed within kernels. Therefore, this technique can be implemented in the CNO's top level to (1) obtain a unified global view of the current state and (2) use predictive models to support decision making. |

**Table 3.** *Cont.*

| Lit. | Technique/Level | Description |
|---|---|---|
| Liu et al., 2018 [45] | Bayesian fusion/ B and M | This combines multisource data to make inferences. Inferences allow the information extracted to be analyzed. This technique can fuse data from sensors, RFID, weather and images, which are collected at different time intervals. The fusion of these data can generate knowledge. Thus, the CNO's participants can perform product management, monitoring, supply control, object detection, etc. |
| Giese et al., 2018 [46] | ANN/T | A CNO has cold data entry as hot data, and the output corresponds to the data fusion process. From both types of data, Artificial Neural Networks (ANN) can issue adequate forecasts. The ANN can indicate the expected trend of both a product or a CNO's participant. As an example, the number of products that will be returned in a closed-loop SC may be estimated. |
| Khaleghi et al., 2014 [47] | Random set theory/T | Random set theory is known for a high computational and representational power. It can perform single-target Bayes filtering using generalized measurements that cannot be expressed as vectors. This technique can combine unstructured data sources from the CNO such as social networks, portals and text documents. Thus, participants can obtain new knowledge that supports decision making. |

*3.5. Big Data Assurance Dimension*

A CNO that tries to capture the value of big data needs guidelines based on varied techniques. Without clear guidelines, companies will face challenges in using data as a business driver and will be exposed to other risks and costs [48]. Therefore, a framework may be obsolete for a CNO if: (1) it does not enable the six Vs and/or (2) it only supports one big data technique. This dimension allows the building of an ecosystem aware of big data.

The sub-dimension six Vs validates the objective of big data, that is, the acquisition, processing and the analysis of datasets that comply with the six vs. Each V is marked in a checklist when the ecosystem fulfills its purpose.

Additionally, the CNO-BD framework extends this dimension through the sub-dimension Techniques guide, which allows the big data ecosystem to be extended. This sub-dimension provides guidelines of techniques to analyze big data in the CNO; see Table 4. Although the way each technique can be used in each type of CNO is indicated, this does not mean that the technique is limited to such manifestations. Additionally, it is not an exhaustive list. Therefore, there are more applications of each technique in CNOs. These techniques are used in the programmable module dimension.

**Table 4.** Big data analysis techniques oriented to the CNO.

| Technique | Description and Models/Algorithms | Example CNO Applications |
|---|---|---|
| Classification | This classifies data by comparing a set of data that was previously categorized. It is based on supervised learning. Models/algorithms: Logistic regression algorithms. Support vector machine. C4.5. | * Determination of the probability of success or failure to form a VE or a VO in a VBE.<br>* Measurement of the behavior of a participant in a PVC. |
| Association rule learning | This is known as market basket analysis. It is based on association rules to identify the relevance of frequent relationships. Models/algorithms: Association rule mining algorithm. Eclat algorithm. | * Determination of the type of product and which channel it is returned by in a closed-loop SC.<br>* Analysis of biological data to obtain new associations through a VL. |
| Statistical modeling | Although this is useful in a variety of problems, it is typically used to determine forecasts and trends. Models/algorithms: Linear regression algorithms. Markov models. Time series. | * Determination of the effect of a new policy on a VG established by external regulatory frameworks.<br>* Estimation of status or condition of the equipment that uses IoT in Industry 4.0 (VE). |

**Table 4.** *Cont.*

| Technique | Description and Models/Algorithms | Example CNO Applications |
|---|---|---|
| Cluster analysis | This classifies input data into different small groups according to the degree of similarity. It is based on unsupervised learning. Models/algorithms: K-means clustering. Hierarchical clustering. | * Management of user profile in an EE.<br>* Marketing through the segmentation of customers in a DVO. |
| Data mining | This integrates statistical and machine learning techniques to discover patterns in large datasets. Models/algorithms: a priori. Pattern matching. Clustering. | * Analysis of text documents for scientific purposes in a PVC.<br>* Processing of food images to determine their quality in a SC. |
| Genetic algorithms | These find true or approximate solutions to optimization and search problems. They are based on the process of natural evolution or survival of the fittest. Models/algorithms: Selection. Mutation. Crossover. | * Computer-aided molecular design for the treatment of diseases (e-Science).<br>* Generation of products and sales campaigns for specific clients (VT oriented to marketing). |
| Natural language processing | This is based on artificial intelligence to analyze human's natural language. It uses soft data. Models/algorithms: Part-of-speech tagging. Sentiment analysis. Speech recognition. | * Determination of the polarity (+, − or neutral) of some aspect in the VG according to social networks.<br>* Analysis of contracts in an insurance agency (DVO) to detect fraud.<br>* Analysis of audio calls of a customer service center of an EE. |
| Machine learning | This predicts a dependent variable (target) based on large known datasets. It creates artificial intelligence using statistical methods and inductive learning. Models/algorithms: Expectation maximization. Deep learning. Principal component analysis. | * Analysis of financial feelings using social networks of a VE.<br>* Analysis of the closed circuit (video) of a VG for perimeter security.<br>* Prediction of readmission rates of patients in a hospital (VO). |
| Crowdsourcing | People, in general, contribute to the accomplishment of a task. Therefore, it is a massive collaboration. It uses multiple formats and types of data. Models/algorithms: Knowledge discovery and management. Distributed human intelligence tasking. Broadcast search. | * Creation of a Kaizen system based on the report of the participants in a VE.<br>* Open call from Disaster Rescue Networks (VBE) to help in a region impacted by a natural disaster.<br>* A massive collaboration of a VL to solve empirical problems. |

### 3.6. Programmable Modules Dimension

This dimension performs intelligent processing of big data through three programmable modules or components. These components are located at the CNO's architecture levels (see Figure 2). Programmable modules mainly use the participant's local data. Therefore, the modules require low-end infrastructure unlike the DPC. Figure 3 shows the interrelation of the three programmable modules described below:

- Data Fusion Module (DFM): This module takes advantage of the CNO participant's local data for two main purposes: The first is the preprocessing of data through the approaches: data alignment, compression, decentralized or inclusive (see the approaches dimension). Preprocessing fulfills the participant's requirements. The second is data fusion based on the techniques shown in Table 3. Participants' requests are resolved without consulting a higher level in most cases. Thus, data fusion uses the information of both the other two modules and the participants. If there is no success, the DFM notifies the Learning Module (LM), which takes control and decides on the proper processing approach. On the other hand, data fusion can be performed globally in the DPC; this process depends on the LM's decisions.

- Learning Module (LM): This module has two main functions. First, it collects, mines and exploits the knowledge generated in a CNO. LM implements the big data analysis techniques shown in Table 5. It monitors and evaluates the information received or issued by each participant. Additionally, it generates information and patterns from incoming heterogeneous sources and output behavior. The former may be used in big data analysis types: descriptive, predictive and prescriptive. The second function of the LM happens after the DFM passes the processing control to LM. This is because local information is not enough to meet the participant's request. Then, LM decides which of the two approaches proceed according to the learning and patterns

generated: centralized or inclusive; see the approaches dimension. Regardless of the type of processing approach, the resulting information is replicated and given to the requesting participant.

- Cache Module (CM): This module reduces the number of requests of the next higher level and/or of the DPC by using cache. Cache allows each CNO's participant to register and use both: (1) the most popular valuable information of the CNO and (2) metadata embedded in big data fragments that describe their attributes and characteristics. The acquired cache is quickly classified, compressed and adapted; this maximizes CM's capabilities for big data and CNO's dynamism. CM can substitute information based on cold data for information based on hot data by means of a data structure that orders the information based on its temperature. As a result, CM has the most popular information of the CNO. Additionally, the CM considers cache in real time, which allows information to be obtained on the fly through interoperability between participants. Thus, CM can answer some participant's requests without consulting the superior level. Hence, CM can reduce the consumption of resources of the CNO.

**Table 5.** Dimensions of the interoperability IRIS framework.

| Dimension | Question Responded | Description of Dimension |
|---|---|---|
| Methodology | How is the measurement performed? | It systematically specifies the activities, tasks, resources and roles necessary to achieve EI. |
| Models and modeling languages | What models and modeling languages are used? | It is based on three approaches: (1) enterprise models (I*, UML 2.0 and BPMN); (2) model-driven interoperability (model-driven approaches composed of the CIM, PIM and PSM models); and (3) model morphisms. Additionally, it uses the MDKe-IRIS modeling language. |
| Techniques | What mechanisms are used to collect and record project information? | It uses three techniques to support the methodology: open-ended interviews, collaborative work sessions and templates. |
| Interoperability measurement system | How are problems and opportunities to improve EI identified? | It measures the potential interoperability through the MM-IRIS maturity model. It has these interoperability views: business, process management, knowledge, human resources, ICT and semantics. |
| Semantic alignment | How are the syntactic and semantic drawbacks of the domain solved? | It uses specific ontologies for each participant. Ontologies are based on the general-purpose thesaurus of the domain where the framework will be applied. |
| Technologies | What technologies are used? | It uses TAOM4E, agent-oriented software development, case tools for BPMN modeling, Olivanova, Eclipse, Java, Protégé, SOA and Incremental Commitment Model to define the testing strategy. |

### 3.7. Interoperability Dimension

This dimension harmonizes the big data ecosystem through EI. EI allows the removal of incompatibilities between the CNO's participants. Thus, each participant keeps its focus on and autonomy in processes, technologies, terminology and operational philosophy. This dimension has three essential aspects: (1) the measurement of interoperability potential to analyze participants' AS-IS situation, (2) a maturity model that establishes levels of interoperability through quantitative values and (3) a methodology that orders the steps and factors necessary to achieve EI.

The CNO-BD framework uses the Interoperability IRIS framework [13] which covers the three essential aspects mentioned above, utilizing the six dimensions described in Table 5.

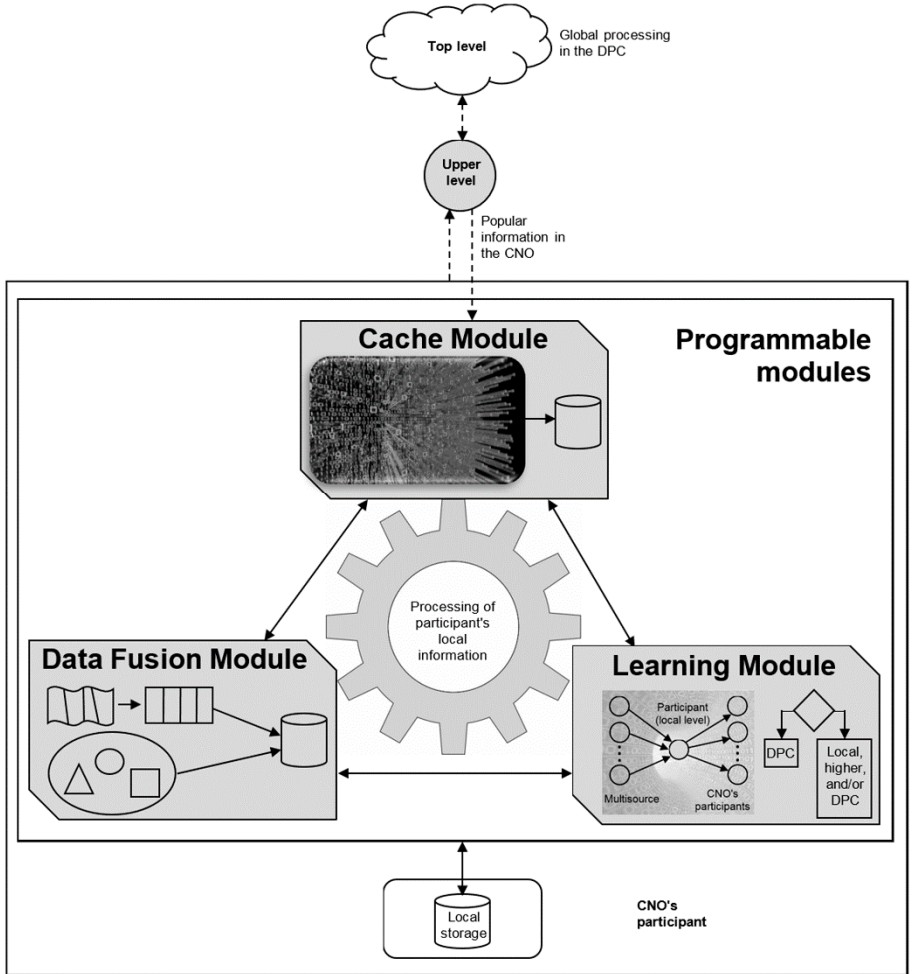

**Figure 3.** Intelligent processing of big data through programmable modules.

The Interoperability IRIS framework not only covers the technical aspects required to achieve EI, but also includes:

- improvement strategies in the short and medium term (TO-BE situation);
- human resources' view;
- the flexibility of use in scenarios of collaboration other than business such as VT, disaster rescue networks and PVC.

## 4. Framework Validation

The first version of the CNO-BD framework considered the scientific and academic advances that address this line of research, in addition to the experience acquired by the authors in the development and implementation of big data ecosystems. The next step was to improve and validate the CNO-BD framework using the expert assessment and study case methods.

### 4.1. Expert Assessment

The CNO-BD framework was evaluated by five expert professionals who have extensive knowledge and experience in big data projects and belong to different companies or institutions. This qualitative evaluation began with an explanation of the CNO-BD framework to the experts. Afterward, both individual and semi-structured interviews were conducted to obtain feedback. In the interviews, open questions were asked according to the responses given. Additionally, predetermined questions were also asked to deepen in relevant aspects. The experts made an objective criticism of the CNO-BD framework

based on the explanation of the modules and scopes exposed, not because they used the framework.

The information collected in the interviews confirmed the viability of the CNO-BD framework as a support tool in the building of a big data ecosystem oriented to the CNO. Although some points were unfavorable, the clarity, relevance, generality and degree of coverage of the CNO-BD framework were recognized.

The first version of the CNO-BD framework was updated considering all the observations and suggestions made by the experts. The update of the CNO-BD framework involved the aggregation of new dimensions with their respective sub-dimensions.

*4.2. Case Study*

This section describes a case study in which the updated CNO-BD framework was applied to (1) test its practical usefulness, generality and completeness and (2) improve it based on the feedback obtained from the CNO's participants and the resulting conclusions.

Our case study was focused on the collaboration of different participants belonging to a CNO of the education sector in Mexico. This study was carried out for 11 months and is based on the work plan developed by [49], which consist of four phases described below: case study selection, understanding the project, data collection and data analysis.

4.2.1. Case Study Selection

The profiles of the three CNOs' participants involved in the case study are described below:

- Alpha University: This is a university that teaches undergraduate and graduate students and offers several academic programs. It periodically delivers results to the Admin Pub. Furthermore, it generates information related to educational indicators, student profiles, approval and dropout rates. Alpha University is a pseudonym used to refer to a CNO's participant in this case study. Similarly, pseudonyms are used for other participants involved in the case study to preserve confidentiality.
- Admin Pub: This is the public administration that governs the state belonging to the country of Mexico. It issues information on events, achievements and areas of opportunity related to educational and social aspects. Although the Alpha University and the state work under the direction of the Admin Pub. The Admin Pub also has a social commitment with the Society Group to ensure the growth and well-being of the people.
- Society Group: This is a group of interrelated people in a common context in time and space. They often share belief systems and economic, ideological, political or educational philosophies. Society Group issues information on their degree of satisfaction with educational aspects that have been achieved or failed.

The CNO was selected because Alpha University was interested in exploiting both the local information that it generates and the isolated information issued by the Admin Pub and the Society Group. Alpha University also tried to detect areas of opportunity using both cold data and hot data. Thus, the CNO's participants could take advantage of both the historical information and the information they generate daily. Additionally, the CNO was interested in storing, processing and visualizing information combined with that of other participants locally or globally according to the available infrastructure. Thus, participants could access big data in an agile and timely manner.

4.2.2. Understanding the Project

The unit of analysis was a collaborative scenario. This involved identifying and managing a host CNO and the information of processes carried out among its participants. The project activities followed this sequence: (1) the measurement of interoperability; see the interoperability dimension. Five interoperability views (business, process management, knowledge, human resources, ICT and semantics) were considered in the maturity model. (2) After having understood the flow and type of information generated by CNO's participants and their infrastructure capabilities, it was determined that Admin Pub would be

in the top level, Alpha University in the middle level, and Society Group in the bottom level; the three levels were designated according to the levels dimension. The mentioned architecture allows associating components of the rest of the dimensions. (3) The data were collected and analyzed. To achieve this, different requirements were considered following the recommendations of the approaches dimension. (4) The functional, technological and graphic design of a system were assessed to evaluate the relationships and patterns in the educational context of the CNO. (5) The system was developed considering approaches and characteristics not contemplated in the typical collaboration among the participants, such as students' profiles at the beginning of their studies of whom have graduated, degree of satisfaction, trend identification or areas of opportunity. Technologies such as JMS and AMQP were applied for the acquisition of data from the sources included in the data source dimension. Additionally, advanced techniques explained in the data fusion dimension and the big data assurance dimension, such as Bayesian fusion, artificial neural networks, natural language processing and data mining, were implemented in the components of the programmable modules dimension to analyze and process big data. Additionally, the Amazon Elastic Compute Cloud (EC2) was used to obtain indicator values. Additionally, 12 processes were modeled and updated to align them to the mentioned system. To achieve this, the IDEF0 and BPMN modeling languages and techniques proposed in the Interoperability IRIS framework were employed. (6) After customizing the system, it was installed and tested. Respective user manuals were prepared. Thus, the big data ecosystem oriented to the CNO was built and implemented. (7) Finally, monitoring and control tasks were scheduled.

### 4.2.3. Data Collection

Qualitative data were collected using three methods: (1) observation, which was a primary source in this study; the field researchers essentially collected data on activities carried out by the CNO's participants, including the data they generate in the processes; (2) semi-structured interviews, the purpose of which is to gather information from a guide. The guide is composed of open questions and other questions planned to maximize coverage; and (3) documentary sources of data, such as reports and files used by the CNO's participants.

### 4.2.4. Data Analysis

An example use case of the CNO-BD framework being used to build the big data ecosystem oriented to the CNO is provided in Table 6. The first two columns describe the use case indicating the target knowledge desired by the CNO and the specific objectives according to the required analysis.

The CNO's participants were located based on the levels dimension. The data sources dimension provides varied data sources, both private and public, for example, SQL Server, CSV, PDF, Facebook, Twitter, HTML documents and word processing documents. These input sources are obtained through technologies and acquisition methods. Additionally, sources were combined to extract only the relevant values for the obtain knowledge. Thus, only the texts of social networks that have an association with the CNO's participants or that belong to their geographical area are filtered. The centralized, decentralized and inclusive approaches—see the approaches dimension—allows the intelligent processing of big data according to the needs and capabilities of the CNO's participants. Interoperability views—see the interoperability dimension—indicate the business layers that should be considered to remove incompatibilities that obstruct collaboration. The big data ecosystem uses the following interfaces in the use case: sheet, SQL, script, graphic and visual. The DFM, LM and CM of the programmable module dimension implement techniques of the data fusion dimension and the big data assurance dimension. Furthermore, Amazon EC2 and free distribution software were used to intelligently process big data.

**Table 6.** Use case of the big data ecosystem oriented to the CNO.

| Use Case | | | Ecosystem's Modules and Components | | | | | |
|---|---|---|---|---|---|---|---|---|
| Target Knowledge | Objective | CNO's Participant | Data Source | Input Variable Format | Approach | Intero-perability View | Interface | Programmable Module |
| Students' profiles at the beginning of their studies of whom have graduated | DE | AU AP SG | ST (SQL Server) SS (CSV, PDF) UN (Facebook API, Twitter API) | RDBMS Text published on social networks | CE | PM KN IS | SQL Script | DFM, LM, CM (EC2, free distribution software) |
| Degree of satisfaction | DE | AU AP SG | ST (SQL Server) SS (JSON, HTML, XML) UN (Facebook API, Twitter API, Word processing documents, PDF) | Documents Text published on social networks | CE DC | BU KN HR IS | Sheet Script Graphic | DFM, LM, CM (EC2, free distribution software) |
| Comparison of trends | PR | AU AP SG | ST (SQL Server) SS (JSON, HTML, XML) UN (Facebook API, Twitter API, Word processing documents, PDF) | RDBMS Documents Text published on social networks | IN | BU PM KN IS | SQL Sheet Script Visual | DFM, LM, CM (EC2, free distribution software) |

Target knowledge helped improve three business processes. This improvement is based on (1) the AS-IS and TO-BE situations of the interoperability dimension, (2) quantitative parameters obtained from the maturity model of the same dimension and (3) techniques and good practices that allow us to take advantage of big data in the CNO.

Those involved in the CNO who participated in the project said they were satisfied with the results of the big data ecosystem. They indicated that the CNO-BD framework supported them for the building of the big data ecosystem because it integrates the main components that should be considered. Additionally, they indicated that the CNO-BD framework is contextualized naturally in a CNO because it recognizes both technological and nontechnological aspects. For instance, the interoperability dimension considers the views of business, process management, knowledge, human resources, ICT and semantics. Almost all the feedback and observations were in favor of the CNO-BD framework. Therefore, the participants indicated that guides of the techniques served as a reference for possible applications in their context and that the levels—see the levels dimension—allowed us to use a virtual hierarchy to take advantage of big data aligned to its purposes as CNO.

In contrast, the few disadvantages observed served to improve the CNO-BD framework. Some of these disadvantages were the lack of a methodology to guide the activities derived from the framework and absence of privacy and security policies that guarantee the entry and exit of data among the CNO's participants. Participants became more interested in the topic because they noticed that their isolated data sources can generate knowledge when these are combined, processed and analyzed with the tools, methods and big data techniques appropriate to the CNO's context. Additionally, once they noticed the competitive advantages offered by big data, they started to sketch a roadmap, with new projects being described that can take advantage of the CNO-BD framework.

Therefore, the CNO-BD framework is an effective reference to build a big data ecosystem to KM of the CNO. The former helped to overcome the CNO's established objectives.

After concluding with the work plan, the CNO-BD framework was updated by considering the feedback and areas of opportunity discovered.

## 5. Discussion

### 5.1. Contributions to Theory

Big data allows CNOs to increase their competitiveness by improving their strategy, management and business processes. Big data has recently emerged in CNOs and forces them to deal with different organizational, technological and human challenges.

Ten big data frameworks have been proposed to deal with these issues in a satisfactory way. Based on the review of these frameworks, a metaframework has been developed. A metaframework is an abstraction of a framework, highlighting the properties of the framework itself. The initial metaframework identified six dimensions and several sub-dimensions addressing different important aspects that a proper big data framework for CNOs should fulfill. Then, during the development of the CNO-BD framework, it was noticed that although the integration of big data in the companies is recognized as a key success factor of a big data project [50], none of the existing frameworks consider the management implications in big data projects. Moreover, neither do they explain when and how to improve the CNO's strategy or business process using big data. As a result, big opportunities for business performance improvements can be lost. To deal with these issues, a new dimension (*Programmable Modules*) and some sub-dimensions are necessary to add to the initial metaframework. Therefore, the metaframework is the first contribution of this study. Now, it is possible to analyze the scope, features, strengths and weakness of existing and future frameworks for big data ecosystems development in CNOs, mapping them with the metaframework.

The second contribution of this study is the analysis of the ten existing frameworks for big data ecosystems development in CNOs using the metaframework (Table 1). The main conclusion is that although existing frameworks consider several dimensions, none of them include all the dimensions and sub-dimensions that are needed to properly support practitioners. Therefore, by using only one of these frameworks some important questions are always missing. Academics interested in improving theses frameworks can see the in Table 1 the areas of improvement for every framework.

The analysis of existing frameworks proved that they have some weakness. For this reason, there has been a need to develop a new framework structured in seven dimensions: the CNO-BD framework. This is the third contribution of this study. In its development, it has been taking into account one of major threats to the potential benefits of big data in the CNOs: A framework for big data ecosystems implementation in CNOs should not consider only technological aspects. This could imply:

- obtaining unfavorable results such as gaps related to interoperability analysis of the organizational structure, business processes or services [51];
- inadequate guides for the use of the different big data techniques [52];
- a lack of ability to filter out irrelevant information to reduce unnecessary costs [53];
- suboptimal handling of data connections [52];
- a lack of contextual information [54];
- a lack of adaptation to approaches according to requirements, that is, a lack of intelligent processing of big data [54].

Considering this background, the proposed CNO-BD framework offers two main novelties and contributions: *First*, it covers all the dimensions and sub-dimensions that a proper framework for big data ecosystem development in CNOs should have, offering solutions for both essential technological issues such as the necessity of addressing interoperability [12] or the processing of heterogeneous data [23], and the linkage with CNO strategy and business process (see Figure 1). Therefore, it allows obtaining meaningful and organized knowledge from the CNO's participants big data to support CNO decision making. *Second*, unlike the existing frameworks that are developed for a specific CNO manifestation, the CNO-BD framework can be used by the different kinds of CNO's identified in the taxonomy developed by [10]. Moreover, in contrast to previous frameworks, the proposed CNO-BD framework addresses the main aspects of big data in a more structured and

complete way. Thus, it is easier to understand for both researchers and professionals and it is unnecessary to train employees in special computer skills or systems administration.

The clarity, relevance, generality, practical usefulness and degree of coverage of the CNO-BD framework were validated by two methods: first, by big data experts in and second, by its use in a case study which yielded favorable conclusions of the participants involved in the project. The case study allowed to use real-life data to generate knowledge.

At this point, it is important to remark on the relation of the CNO-BD framework with the topic of Enterprise Architecture (EA). EA is a discipline that provides a set of principles, methods, models and tools to provide a vision of all companies existing systems and allows the alignment of decisions with the existing data [55]. The main elements of EA are methodology, framework and modeling language [56]. The CNO-BD framework would be a specialized framework for big data in CNOs inside a generic EA proposal. Moreover, with the programmable modules dimension this framework avoids one of the most common worst EA practices: no linkage to business strategy and targeted business outcomes [57].

### 5.2. Implications and Suggestions for Practice

With the CNO-BD framework, the CNO can combine heterogeneous data sources, as well as to have clear guidelines to overcome the challenges posed by the large volumes of data generated in their participants' collaboration. Due to the competitive benefits offered by big data, the CNO will continue making efforts to take advantage of it. In doing so, the results of our study suggest that the CNO must: (1) locate each participant within a virtual architecture (levels); (2) learn to value the great potential that their heterogeneous data sources can generate; and (3) prepare roadmaps to direct and measure their development.

The technical aspects of the case study have been solved with the use of Amazon EC2 and free distribution software to collect, store, represent and disseminate data from different sources and formats. This software simplifies the building of the big data ecosystem without depending on the chosen approaches for preprocessing. The latter is an advantage for the CNOs because big data can be processed locally, globally (DPC) or both (inclusive processing).

### 5.3. Limitations and Future Research

Due to the novelty of the topic, our study considered several fields of research, and its evaluation resulting from the CNO cannot be considered definitive.

On the one hand, the success of the big data ecosystem is affected by the execution of the different activities derived from the CNO-BD framework. Most of the activities are sequential, with the possibility of applying control loops that allow returning to previous activities to refine the process. An ordered and structured execution of the activities influences the possible results obtained from the intelligent processing of big data. Furthermore, there are additional activities that may be required, such as processing and disseminating sensitive data under privacy and security policies [58]; and the relation and inclusion of other emerging technologies, such as the Internet of Things or cloud computing, could be analyzed [59]. In this sense, the CNO field may be interested in future research projects that propose a methodology to guide the execution of different activities. Therefore, in future work, we should consider the development of a methodology or even extend the CNO-BD framework to include aspects of privacy and security.

Finally, the CNO-BD framework was used only in one CNO. Therefore, this needs to be validated by different CNOs' manifestations. Thus, other aspects of big data could be detected to strengthen the CNO-BD framework.

## 6. Conclusions

Big data generated in the CNO must be analyzed together. However, the complexity of combining CNO's heterogeneous data has been continuously increasing. To face this challenge, frameworks that help in the analysis and processing of big data have been reported in the literature. These frameworks were analyzed using the metaframework developed, showing that they use a single big data technique, consider a single CNO's manifestation, do not provide a logical design of the CNO's levels and do not enable the intelligent processing of big data.

To overcome the weaknesses of the existing frameworks, the CNO-BD framework for big data ecosystem development in CNOs was developed. The CNO-BD framework was developed considering the scientific and academic advances that address this line of research, and the experience acquired by the authors in the development and implementation of big data ecosystems. The framework was validated both by experts and by a case study. The case study showed that both the building of the big data ecosystem and the intelligent processing of big data in CNOs are effective. Participants in the case study corroborated that the use of the CNO-BD framework provided support to collect, store, represent, and disseminate data from different sources and formats.

Regarding the scope of future work, there are different proposals, such as (1) the development of a methodology that guides in the framework application, (2) improving the framework considering aspects of privacy and security and (3) applying the framework to different CNO's manifestations.

The findings of this paper are also valuable for researchers and professionals who develop projects which aim to take advantage of big data in associations, strategic alliances, inter-organizational relationships, coalitions, cooperative or collaborative arrangements, or who develop frameworks that help to test big data theories within the CNO.

**Author Contributions:** Conceptualization, J.-A.H.-A. and R.C.; investigation, J.-A.H.-A.; methodology, J.-A.H.-A. and R.C.; supervision, R.C., R.V.R.-H. and R.M.-C.; validation, R.V.R.-H. and R.M.-C.; writing—original draft, J.-A.H.-A. and R.C. All authors have read and agreed to the published version of the manuscript.

**Funding:** This research received no external funding.

**Institutional Review Board Statement:** Not applicable.

**Informed Consent Statement:** Not applicable.

**Conflicts of Interest:** The authors declare no conflict of interest.

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
