# Peer review of "A Framework to Build a Big Data Ecosystem Oriented to the Collaborative Networked Organization"

_applsci, doi:10.3390/app122211494_

Round 1

Reviewer 1 Report

The contribution of the manuscript is the interoperability method of Collaborative Network Organization.

The article is structured correctly and the content is presented in a logically consistent order.

However, the "Abstract" section lacks at least mentioning methods used and a summary of the results. The authors should refine that section thoroughly. Authors should convey the essence of their approach in a concise manner.

The authors did not turn on line numbering. This makes it difficult to relate to individual sentences in the text.

References to items in the literature should be formatted by the authors in accordance with the MDPI requirements. For example, the fragment in line 4 of the "Introduction" section should be "Cotrino et al. [23]". It is also a good practice to include bibliographic items in the list of references in the order they are cited in the text of the article.

In the "Discussion" section, the authors should discuss the pros and cons of their approach. In my view, one aspect is missing. The authors have not mentioned the topic of Enterprise Architecture (EA). Collaborative Network Organization can be seen as a complex corporation that comprises many companies but also cooperates with external ones. Generally, I would like the authors to comment on their framework and EA in the broader context. For example: How can the approach, presented in the manuscript, be used in EA? I recommend using in the discussion two papers presenting examples of EA approaches from various industries:

- Towards Enterprise Architecture for Capital Group in Energy Sector, https://doi.org/10.1109/INES.2018.8523941

- Enterprise architecture for high flexible and agile company in automotive industry, https://doi.org/10.1016/j.procs.2021.01.303

The "Conclusions" section should be more supported by the results. 

In the "References" section the authors should add DOIs for papers.

The English used in the manuscript requires additional effort to correct as far as the style and punctuation are concerned. It should be noted that the article contains editing errors. They do not detract from the substantive value of the article but should be eliminated from the manuscript. Please check the manuscript once again.

Author Response

Please, see attached file

Reviewer 2 Report

Dear Authors,

              The paper titled “A Framework to Build a Big Data Ecosystem Oriented to the Collaborative Networked Organization” has been taken up for review. The manuscript requires major revisions .

Author Response

Please, see attached file

Round 2

Reviewer 2 Report

Dear Authors,

             The paper titled “A Framework to Build a Big Data Ecosystem Oriented to the Collaborative Networked Organization” has been taken up for review. The topic is interesting and the author response to reviewer comments was satisfactory.
